# An Integrated Model to Improve Job Satisfaction: A Case for a Sustainable Construction Industry

Qasim Hussain Khahro [1,*], Noor Yasmin Zainun [1], Shabir Hussain Khahro [2,3,*] and Basel Sultan [2]

1. Jamilus Research Center, Faculty of Civil Engineering & Environmental Engineering, Universiti Tun Hussein Onn Malaysia, Parit Raja 86400, Johor, Malaysia
2. Department of Engineering Management, College of Engineering, Prince Sultan University, Riyadh 11586, Saudi Arabia; basel.sultan@psu.edu.sa
3. Educational Research Lab, Prince Sultan University, Riyadh 11586, Saudi Arabia
* Correspondence: qhkhahro13@gmail.com (Q.H.K.); shkhahro@psu.edu.sa (S.H.K.)

**Abstract:** In the last few years, the fields of management, social psychology, and business operations have all paid a large amount of attention to the academic idea of job satisfaction. This paper looks at more than a decade of research into what makes people happy at work and what happens to them as a result. Companies have started to realize that their employees are their most valuable asset in this time of rapid change. There is no specific model of the job satisfaction of construction workers in developing countries. Thus, this paper evaluates the different job satisfaction models and also proposes an integrated job satisfaction model for construction industry workers. The data were collected from experts in the construction industry using a questionnaire survey and almost 290 experts participated in this research to assist in the development of the model. The data were analyzed using SPSS. The model was developed and validated using Smart PLS. Eight key aspects were found to be very important to improving the job satisfaction of construction workers in developing countries. Job compensation packages given to construction workers, work–life balance, career growth, and job security are the top key features investigated in this study. It is concluded that satisfied and motivated employees are imperative for the construction business, and that this is also a key factor that separates successful companies from others. The findings of this paper contribute to UN-SDG 8 "Decent Work and Economic Growth" and UN-SDG 9 "Industry, Innovation, and Infrastructure".

**Keywords:** job satisfaction; job performance; job productivity; job culture; construction industry

## 1. Introduction

Since the beginning of the academic study of organizations, one of the primary focuses of organizational scholars has been to gain an understanding of the factors that contribute to job satisfaction. In the most recent few years, a significant number of these researchers have shifted their focus to the part that organizational justice plays in determining this fundamentally important aspect of work attitude. The question of whether there is a surface link or a genuine relationship between job satisfaction and job performance is a popular subject in the field of organizational behavior, and has received a large amount of attention as a result. Both the academic area and the professional sector have shifted their attention to the importance of having productive and happy workers in the workplace. The study of the association between happy workers and productive workers has gone through a phase of shifting from a positive correlation to a negative correlation. There has been a growing volume of research in recent years that supports the idea that high levels of emotional intelligence (EI) are a strong indicator of professional success. Attitude outcomes, including teachers' work satisfaction, burnout, and organizational commitment, have been the primary focus of previous research on the attribute of EI [1].

Organizational justice is the single most significant aspect of any company, and it has been shown that focusing on the links between job satisfaction and justice at work may enhance these types of research [2]. Workers join organizations with a set of expectations and a set of skills they want to acquire (i.e., money, comfort, personal growth, learning, etc.). When a worker's experiences at work often line up with their expectations, that worker is more likely to be content in his or her position. This means that workers' opinions of their jobs are reflected in their levels of satisfaction. Job satisfaction is also thought to be a predictor of whether or not a person would leave their position [3,4]. As a highly capital- and technology-intensive sector, the high-tech sector places a premium on the rate of innovation. Maintaining a competitive edge requires always upgrading and improving technology. Research and development (R&D) workers are essential to a company's success because of their capacity to think creatively and solve problems, but are often in short supply in the high-tech sector [5]. Previous findings showed that worrying about contracting COVID-19 not only directly lowered job satisfaction but also indirectly impacted individuals' levels of psychological capital [6].

Turnover among competent workers is a major issue for businesses throughout the world. Profitability, efficiency, and general quality of output are all influenced by turnover. An employee's decision to stay at or leave a company is heavily influenced by the HR department, which is understandable because HR is often cited as an organization's primary source of sustainable competitive advantage. Turnover is expensive because of the time and effort required to recruit and hire new personnel who have the necessary skills and experience, as well as the time and effort used in training them. Employers must gain insight into employees' decisions to leave or remain with a company since this provides valuable data for formulating retention strategies [7]. Studies have cited diverse reasons for why employees leave their jobs. Most of these issues, however, can be traced back to salary and benefits, employee services, training and development, performance management, and job security, to name a few human resources practices. Companies that rely on the global labor market to maintain their employment levels may face crisis levels of employee turnover. As a result, it can be difficult to keep employees happy and motivated in a global setting. The time and resources invested in finding and hiring candidates from abroad, as well as in providing the necessary training before workers can contribute effectively, can add up to a significant sum for businesses. Therefore, these businesses must hold on to these workers [7].

The topic of 'employee engagement' has been widely studied in the business world. Academic studies have shown a connection between engaged employees and increased levels of corporate commitment and civic conduct. Psychological factors including a sense of purpose, security, and accessibility have been linked to employee engagement in the workplace, according to empirical studies [8]. Work engagement is also connected with essential aspects of work-life, including workload, control, recognition and reward, communication and social support, perceived fairness, valued work, and work that is considered to be valued [9]. Employees who are engaged in their work are more likely to feel a stronger commitment to their company and are less likely to consider leaving their current employer. According to these findings, engagement is an important factor to consider when attempting to forecast employee outcomes, organizational performance, and corporate reputation [10].

Thus, in such a scenario, it is imperative to design a model to improve the job satisfaction level of workers working in the construction industry of developing countries. There is no such model in place to gauge the satisfaction level of construction workers, and the few existing models from other industries are not efficient enough to be used in the construction industry. This is because different dynamics and factors are involved in each industry, including the construction industry. The workers of this important industry are known as deprived workers in terms of benefits, labor laws, and facilities. Thus, the workers are not motivated, and as a result, productivity is being affected. Therefore, this paper investigates the existing models and proposes an integrated model for construction

industry workers to improve their satisfaction level. It has been reported in various studies that satisfaction is directly linked with the productivity of the workers, so this model can assist stakeholders in enhancing the job satisfaction culture in their companies to support construction workers in particular and the industry in general.

## 2. Literature Review

Qualitative and quantitative analysis has been undertaken of the link between work satisfaction and job performance. Qualitative research is structured around seven models that characterize previous studies on the connection between work satisfaction and job performance. In part due to a lack of incorporation and integration in the literature, research has not produced convincing confirmation or rejection of any model, although certain theories have garnered more support than others [11]. As a result of the Hawthorne experiment, the possible association between employee attitudes and job performance, as well as the promotion of the interpersonal relationship movement, gained increased academic interest in the early 1930s. Employee Attitudes and Employee Performance, published by Brayfield and Crockett in the American Psychological Bulletin in the middle of the 1950s, was the most influential assessment of studies at the time [12].

A post-cognitive no-recursive model was tested in which job satisfaction follows job perceptions in the causal order and the two are intertwined; a precognitive recursive model was tested in which job perceptions follow job satisfaction in the causal order, and affect but do not cause job satisfaction; and a precognitive no-recursive model was tested in which job perceptions follow job satisfaction but are not causes of job satisfaction [13]. In the context of modern turnover models, the links between structural drivers of work satisfaction and organizational commitment were examined using meta-analytical structural equation modeling (SEM). Only three structural factors (distributive justice, promoting opportunities, and supervisory support) were directly associated with organizational commitment beyond their influence on work satisfaction [14].

Although it is accepted that a positive organizational culture and strong leadership can help public relations professionals do their jobs well, little research has examined the specific outcomes (such as engagement and trust) and traditional employee outcomes (such as job satisfaction) that such conditions might generate at the individual practitioner level [15].

Role ambiguity, defined as 'individual changes across the membership border of a social system that is initiated by the person,' is the primary focus of employee turnover research. Beginning in 1958, this line of inquiry has produced various theories of the fundamental factors and processes of voluntarism. In contemporary models, work satisfaction (defined as the extent to which employees have a positive emotional orientation toward employment with the firm) and organizational commitment are often seen as intervening factors in the turnover process. Numerous meta-analyses have conclusively shown the empirical association between these two factors and voluntary turnover; hence, they are regarded as crucial components of turnover models [14]. The first basic model was presented by [16], in which job satisfaction is linked with some features of the project that have a positive and negative effect on job satisfaction and organizational commitment. The limitations of the model are that it was one of the earliest models to gauge the satisfaction level and it was at a very basic level. Furthermore, job satisfaction and organizational commitment are linked with only five factors, and similarly, the model is affected by five factors.

This model was extended to analyze the relationship between structural determinants and organizational commitment/job satisfaction, as suggested by Kim in 1996, where job satisfaction comes before organizational commitment and has a positive effect on the overall model, as presented by [17]. The limitations of the model are that, similar to the first phase of the model, it was focused on limited features only, whereas, in today's dynamic world, the industries' practices are different so the model's capability to deal with multiple features is lacking.

Finally, the model was extended to analyze the relationship between structural determinants and organizational commitment/job satisfaction, as suggested by Price and Mueller in 1986. In this model, job satisfaction was linked with organizational commitment and three other features of any business, namely, promotional chances, supervisory support, and distributive justice, as shown in Figure 1.

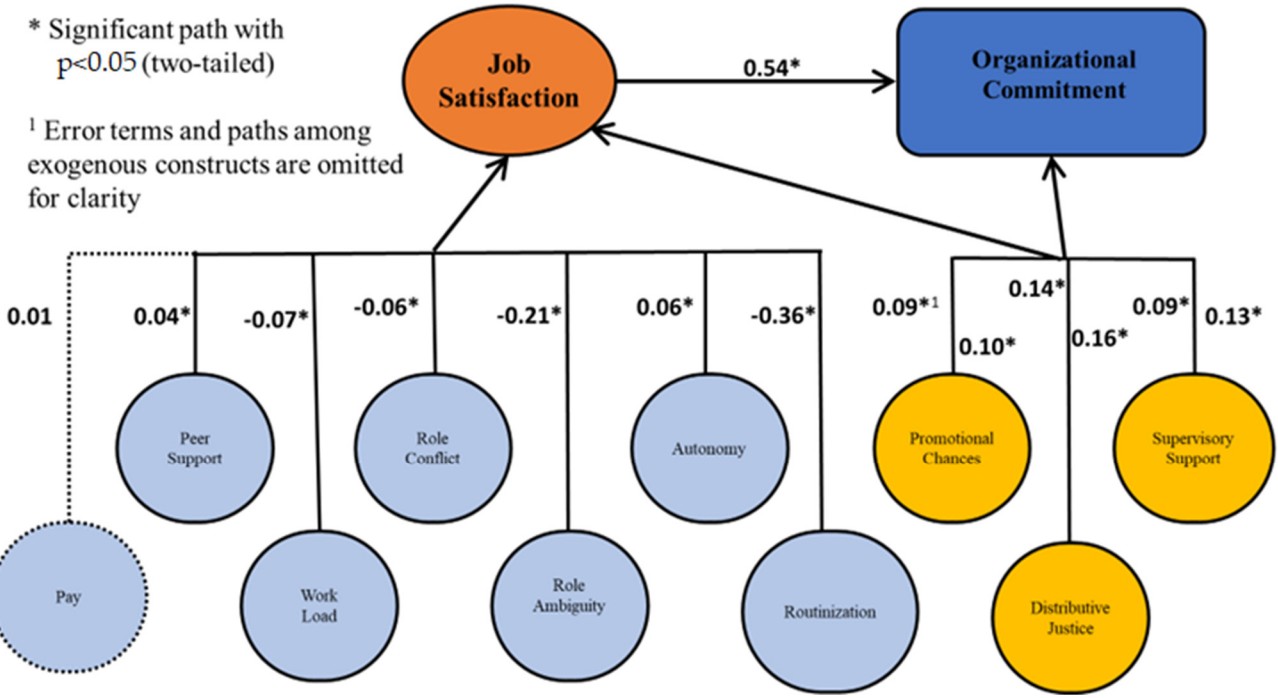

**Figure 1.** Job satisfaction: completely standardized parameter estimates model, modified from [14].

In analyzing the acquired findings of this model, it is important to keep in mind that the sample size underlying the meta-analytical SEM approach is enormous, and far bigger than is often seen in single studies of job satisfaction, organizational commitment, or turnover. As a result, a limitation of this model is that the resultant chi-square statistics are inflated in comparison to previous empirical investigations, and even insignificant associations are statistically significant [17].

Research has examined the moderating effect of national culture on the association between work attributes and job satisfaction. The analytical technique consisted of two independent stages. First, a model of individual-level links between job attributes and job satisfaction within a nation (Level 1) was created. The subsequent step used the within-country regression coefficients as dependent variables to determine how within-country relationships vary as a function of national context (Level 2). For Level 1 analysis, job satisfaction was regressed on job characteristics, with added controls for several demographic variables including age, gender, education, work status (i.e., whether the respondent is in paid work, working as a trainee or apprentice, retired or unemployed, etc.), and work sectors (i.e., private or public). As depicted by [18], for Level 2 analysis, the within-country coefficient estimates were regressed on a country-level measure of culture, namely, national cultural scores.

The limitations of the model are as follows. First, the ISSP's single-item scale was used to gauge employee contentment with their current position. When the construct being measured is multifaceted and complex, it may be difficult for a single item to achieve sufficient dependability [19]. Multiple-item assessments, such as the Michigan Organizational Assessment Questionnaire's three-point assessment of global work satisfaction, can be used in future research. Second, only a subset of the employment characteristics that are measured by the International Social Survey Program (ISSP) was examined. This could

not include all of the factors that are vital for people to have successful results on the job. Physical working circumstances, feedback and social support received, and task diversity are only a few of the other job aspects that may be studied to provide a more complete picture of the impact of working conditions on employee satisfaction.

It is clear from the results of this study that there is a correlation between these two factors in the workplace. The strength of the link between organizational loyalty and work contentment ranges from moderate to strong. Involvement in one's work was found to have a moderating effect between job contentment and loyalty to one's company. Although there is some mediation between job satisfaction and organizational commitment via engagement, the effect of job satisfaction on organizational commitment is largely accomplished directly. The architecture of organizational policies and procedures does not significantly affect employees' levels of satisfaction. It was observed that job features do not play a major role in determining whether or not an employee will feel a sense of loyalty to his or her employer [19]. The limitations of this model are as follows. It appears from the data that the construct of organizational policies and processes does not significantly influence employee satisfaction. There are two possible reasons for this. The first is that this was the only construct for which an appropriate standard questionnaire was unable to be located, meaning the researchers had to develop their own; as such, future research should focus on refining the construct's operationalization.

In another study [12], the authors presented Organizational Behavior and Personnel Psychology, which is a widely used model of work satisfaction determinants in studies of the causes of job satisfaction. Figure 2 proposes a categorization of the elements affecting job satisfaction into two groups: environmental and personal.

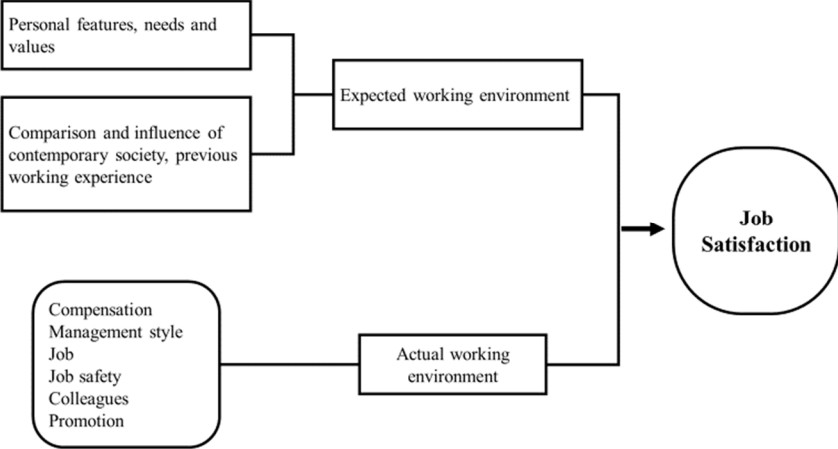

**Figure 2.** Job satisfaction model by Wexley, modified from [12].

The limitations of the model are that organizational culture, business climate, political climate, economic climate, and other external variables are all examples of environmental factors, so each person's unique demographics, skills, personalities, worldviews, etc. should be considered.

Figure 3 displays the research goals, which were to examine the connection between teacher qualities and school working circumstances and their effect on job satisfaction among Swedish eighth-grade mathematics instructors. Teacher workload, teacher collaboration, and student discipline were ranked as the three most significant characteristics of the school environment that contributed to teacher job satisfaction. Job satisfaction was shown to be greater among female educators, educators with more professional development experience, and more effective educators. In addition, male instructors placed a considerably higher value on teacher cooperation, whereas the least effective teachers placed a premium on their judgments of student discipline in the classroom [19].

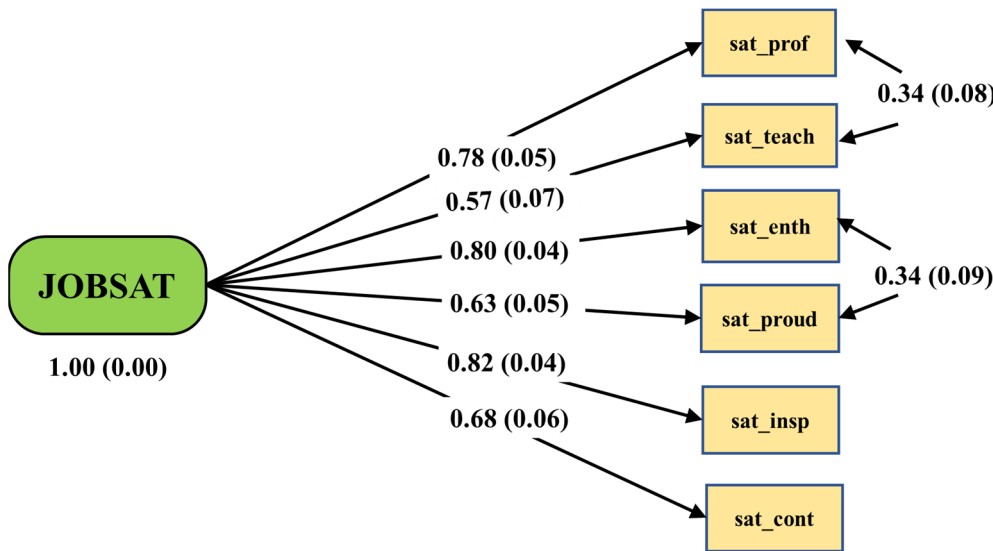

**Figure 3.** A measurement model of teacher job satisfaction. Model modified from [20].

The limitations of the model are that many characteristics of the educational environment could not be considered, although they were included in the Trends in International Mathematics and Science Survey (TIMSS) questionnaire. The results may also be of limited generalizability outside the sample of Swedish eighth-grade mathematics instructors included in this study. Second, further research is needed to better understand the underlying structure of the interactions between school environment elements and different teacher traits. Future research should pay more attention to directionality, as certain connections may be two-way. Lastly, the TIMSS research may not provide causal inferences due to its cross-sectional methodology.

For this reason, a longitudinal study at the national level to track changes in teachers' levels of contentment on the job is necessary. Lastly, comparative research on variables boosting teacher retention is particularly significant since some countries are likely to face large teacher shortages for many decades to come. A job-happiness model is depicted in Figure 4.

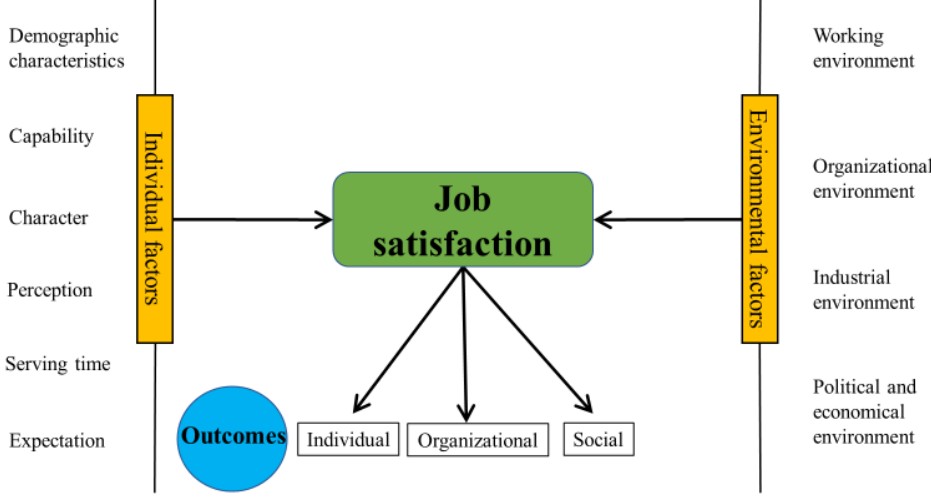

**Figure 4.** Job satisfaction model by Seashore, S. E. and Tobor, T. D; modified from [12].

Another study investigated the connections between dispersed leadership, professional cooperation, and teachers' job satisfaction in American schools using hierarchical linear modeling (HLM). After controlling for other individual and school culture variables,

the results showed that teachers' opinions of dispersed leadership were substantially and positively linked with their work satisfaction [20].

Findings suggest that other characteristics of school culture, such as professional cooperation and self-efficacy, which are positively related to teacher job satisfaction and student accomplishment, are also influenced by distributed leadership, as observed in the theoretical model of teachers' job satisfaction by [21]. Disadvantaged schools have the most to gain from a school design that promotes faculty leadership to create an atmosphere that is favorable to student success. The limitations are that there are several complications in this study. The focus of the study should be on the relationships between the variables because the data are cross-sectional, and concluding cause-and-effect relationships from the study findings should be avoided. The opinions expressed in surveys are snapshots in time that reveal how people feel about their jobs. Longitudinal data from future studies can shed light on teachers' long-term feelings about their jobs and their likelihood of quitting.

A framework for job satisfaction in hospitals was proposed by [22]. The study found no correlation between nurse staffing levels and turnover intentions, job satisfaction, or workplace injuries. The amount of labor involved was directly correlated with both intentions to quit and dissatisfaction with employment. Workload played a significant role in both nurses' intention to leave their jobs and their level of job satisfaction in intensive care units. Research in the future should take place at medium-to-small hospitals located throughout an area. A limitation of the model is that the results may not generalize to other comprehensive nursing care units of small-to-medium size hospitals outside of Seoul and Gyeonggi Province, Republic of Korea, because of the study's geographic limitations. Due to the cross-sectional nature of the research and the reliance on self-reported questionnaire data, it was not possible to draw any firm conclusions about the causal linkages between nurse staffing and nurse outcomes. Future research needs to be longitudinal, examining connections such as whether changes in nurse staffing levels cause shifts in nursing outcomes.

In another study [1], the authors address the mediating function of teacher job satisfaction and the impacts of school-level variables by proposing a multilevel model of the effects of teachers' EI on their work performance. The results showed that teachers' EI favorably affects their work performance, both immediately and indirectly, via their degree of job satisfaction. The results of this research also showed that teachers' EI has an indirect influence on their work performance via job satisfaction and that this effect may be moderated by the amount of organizational trust present within the school. According to the results of the multilevel model, the relationship between teachers' EI and student outcomes was larger in schools with lower levels of organizational trust. Teacher performance was not shown to be correlated with principals' EI.

The findings demonstrated that teachers' EI was positively associated with their work performance and that this link was somewhat mediated by their level of job satisfaction. Teachers' EI was shown to be favorably associated with their work performance, as was their job happiness. Teachers' work satisfaction was shown to have a positive indirect effect on their effectiveness in the classroom, suggesting that EI is a feature that contributes to both. The study demonstrated that organizational trust at the school level acts as a crucial boundary condition for the effects of teachers' EI, suggesting that teachers with greater EI perform better than instructors with lower EI. The results backed up the prediction that teachers' EI is less relevant in predicting job success when there is more organizational trust in schools. On the contrary, when trust in an institution is low, the value of teachers' EI becomes more apparent. In the proposed model of teachers' job satisfaction [1], teacher satisfaction was greater in schools where there was a high degree of organizational trust compared to schools where there was a low level of organizational trust.

The limitations of this model are as follows. First, because of the study's cross-sectional design, it is impossible to know which way the correlations between variables point. Hence, researchers were invited to test the model longitudinally to establish the underlying causal inferences with higher confidence, even when prior research supported the expected

directions of associations. Second, there was inconsistency in the sample size since the sample was drawn from a small number of institutions. This may have had an undesirable impact on the cross-level impacts that were analyzed. Thirdly, there was a problem with the way that many important variables were measured. Teacher self-reports were the only source of data used to operationalize job performance, which may have been influenced by social desirability bias. The model has certain limitations, including the fact that it only accounts for the beneficial effect of teachers' EI on work performance, through job satisfaction as a mediator and organizational trust as a moderator.

Figure 5 also provides a conceptual model showing how factors including wage, training, supervisor assistance, performance evaluation, benefits, employee autonomy, services, job security, working environment, and job design impact employee happiness and loyalty to the company. It is worth noting that workers tend to be more dedicated to their work when they feel appreciated by their employers. Employees' desire to remain in a company is influenced by several factors, including work satisfaction and organizational commitment [7].

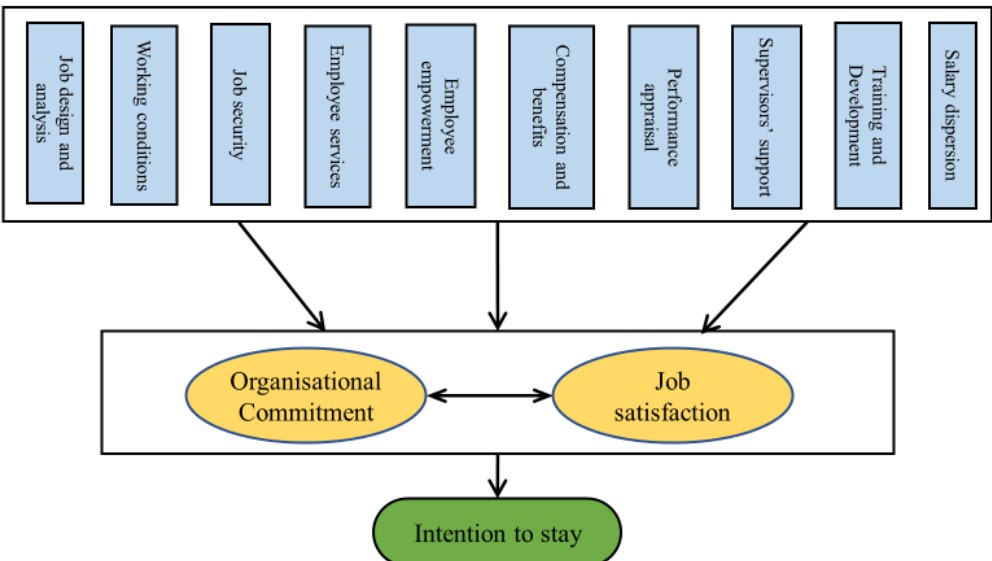

**Figure 5.** Research conceptual model modified from [7].

Several limitations of this research have to be highlighted. The first issue is the size of the sample; while it was sufficient for the number of variables, a bigger sample would improve the reliability of the results. Second, although this study's primary audience was senior officials in the Qatari government, expanding its scope to include NGOs in the country would strengthen the study's overall credibility and applicability. Finally, this study relied on data from a single snapshot in time; however, if the aim is to know why connections exist between variables, this method may not help [7].

The job satisfaction model proposed by [23] shows the results of testing a moderated mediation model that proposes an emotional labor strategy that mediates the relationship between EQ and work satisfaction. It was found that hotel workers' levels of emotional intelligence had a significant bearing on how satisfied workers were with their jobs. It was shown that the connection between emotional intelligence and work satisfaction was moderated by deep acting, but not by surface acting. The research team showed that the relationship between emotional intelligence and work satisfaction was moderated by the perception of organizational support [24,25]. A limitation of this model is that the data in this research were gathered via a cross-sectional questionnaire survey. Cross-sectional studies may be problematic for determining whether or not a model's relationships are causal. The moderated mediation interactions found in this study may be confirmed in future research using various approaches such as a quasi-experimental design. To better

understand the role of emotional intelligence in hospitality management, future research can examine the effects of emotional intelligence on relevant employee attitudes and behaviors, such as workplace well-being perception, turnover behavior, service performance, and organizational citizenship behavior.

Figure 6 displays the expected correlations and the suggested conceptual model, in which engagement and trust operate as joint mediators between a supportive corporate culture, a strong leader performance, and overall work satisfaction. The proposed theoretical framework further shows that involvement acts as a mediator between trust and job fulfillment. The research methodology, construct measurements, and hypothesis testing outcomes are detailed below [15]. According to the theoretical framework, there are robust positive correlations between the observed constructs. While finding no direct beneficial benefits of a supportive organizational culture or a strong leader performance on employee job satisfaction, it is important to understand the powerful intermediary roles of engagement and trust in these connections.

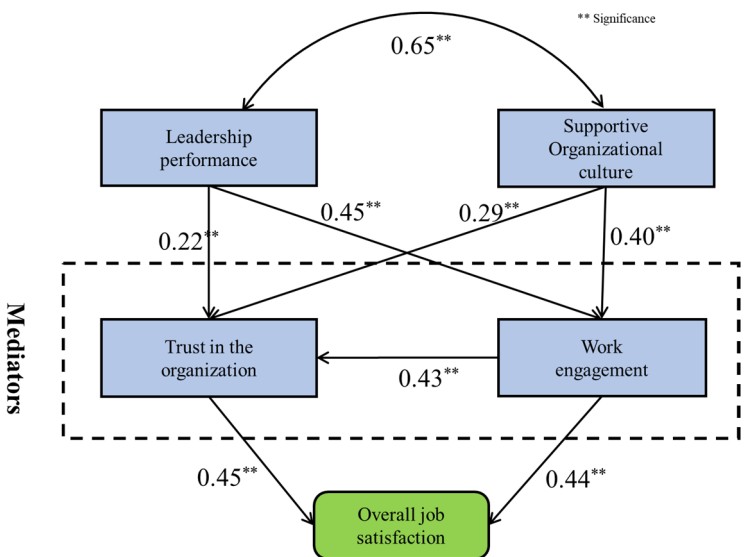

**Figure 6.** Structural model of job satisfaction by engagement and trust; modified from [15].

A limitation of this model is that, although it was able to collect a representative sample of high-level professionals in the field, the study's cross-sectional and self-reported research design makes it vulnerable to common method bias and potentially dampens the strength of the findings regarding organizational culture, leadership, engagement, trust, and job satisfaction. Using a longitudinal study design and a mixed study design (self-report data and observed data) may help researchers shed light on this topic in the future. There may have been cultural barriers since the poll was focused on the United States and public relations practitioners based there. Not enough qualitative background and in-depth insights into the major concepts and linkages were provided. In-depth interviews and other qualitative research methodologies have the potential to shed light on these obstacles and provide more grounded, actionable recommendations for policy and practice. Self-efficacy and impediments to professional growth were mean-centered to facilitate understanding since none of these constructs had a meaningful 0 value. Using the lavaan package in R, a multiple regression analysis was conducted of work satisfaction based on the factors of self-efficacy, impediments to professional growth, and the presence of a mentor [24].

Figure 7 shows that the findings of a model examining teachers' levels of work satisfaction shed light on the link between many key retention factors. For instructors with less than five years of experience, self-efficacy and the availability of a mentor were both favorably connected to work happiness, but impediments to professional growth were adversely related to job satisfaction. Such correlations provide credence to the hypothesis suggesting a link between the two phenomena. Although the theory predicts that self-

efficacy, obstacles, and mentors all contribute to increased work satisfaction, the statistical evidence shown here cannot substantiate the direction of the correlations.

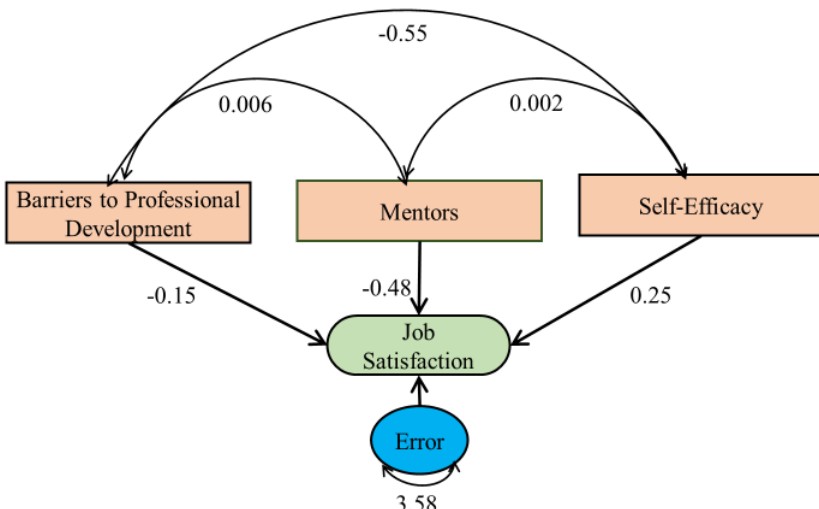

**Figure 7.** Path diagram of final model; modified from [24].

The lack of a variable to account for the demographics of the instructors' workplaces is a limitation of this research. Several studies have highlighted the significance of contextual factors at the school level, suggesting that variables be included in future studies to account for contextual factors such as socioeconomic status and the proportion of children eligible for free and reduced-cost lunch. In addition, the quality of mentorship was not assessed in this research. It is unlikely that just assigning each instructor a mentor will improve things for everyone.

Job satisfaction and engagement were suggested as mediators between resilience and job performance in a model presented by [25]. Combining this theoretical framework with data from across the world might help reveal the correlations between these variables and other settings, which may be unique to the United States. Further study is needed to construct theoretical and statistical models with supporting datasets to better understand work satisfaction since this model only explained 16% of the variation associated with it. Ref. [23] focused on the significance of two factors that contribute to a great work experience: job satisfaction and employee engagement. The study looked at the link between resiliency and productivity at work, and found that it significantly influences the success of those in helping professions. Those who work in assisting professions benefit from a more resilient workforce. Moreover, job satisfaction and engagement, two very good aspects of one's well-being in the workplace, are linked to resilience. Those who can bounce back from adversity more quickly and effectively have higher levels of job satisfaction and participation than their less resilient counterparts. Only involvement at work is linked to actual performance on the job, thus engaging employees to outperform their less involved coworkers, but not the other way around [25].

The limitations of this model are as follows. First, it is crucial to understand where results can and cannot be generalized. As the sample included persons from a wide range of helping occupations, it is believed that the findings are representative of the helping workforce as a whole. Although the results indicated no statistically significant influence on the profession, future studies might benefit from a more representative sample if there is a need to replicate the findings and extrapolate them to all workers [25].

In Ref. [24], authors conducted a study to examine the mediating role of organizational commitment on the link between perceived organizational support, perceived alternative job opportunities, and turnover intention, and the moderating role of job satisfaction on the proposed relationships, as suggested by the SEM [26]. The study examined the structural model using variance-based structural equation modeling (PLS-SEM). Their analysis using

PLS-SEM found that POS has a significant and positive direct effect on organizational commitment (β = 0.71, ρ < 0.000). Perceived alternative job opportunities have a significant and negative direct effect on organizational commitment (β = −0.23, ρ < 0.000). Job satisfaction has a non-significant effect on organizational commitment (β = 0.04, ρ = 0.284). Organizational commitment has a significant and negative direct effect on turnover intention (β = −0.38, ρ < 0.000). It was found that organizational commitment mediates the association between POS and turnover intention. Organizational commitment mediates the association between perceived alternative job opportunities and turnover intention. In addition, job satisfaction does not moderate the associations between organizational support, perceived alternative job opportunities, and organizational commitment.

The limitations of the model are as follows. First, it was limited in its capacity to make firm conclusions about causal inference and to investigate the development of the research variables over time due to the cross-sectional character of the data. A time-lag design might be used in future research. Second, social desirability bias might significantly affect the estimates obtained from self-reports; hence, future research could pool information from a variety of sources. Third, the capacity to extrapolate from this study's setting of Jordanian SMEs is constrained. Continuing studies should use the same paradigm but apply it to new contexts. Finally, future research might include methods such as fuzzy sets, machine learning, and artificial intelligence techniques, even if common method bias (CMB) is accounted for by utilizing procedural approaches. Both good and negative attitudes among workers might be the outcome of their interactions with coworkers and the company as a whole. Prevention efforts may be strengthened by identifying the factors that contribute to an individual's level of organizational commitment and desire to leave a company. Employee turnover is expensive, but this preventative method may help [26].

It is observed through the detailed literature review of previous models that all models are designed for a specific population and condition, and that no such model exists for the construction industry. There is no model in place to measure the job satisfaction of construction industry workers; thus, this paper proposes a job satisfaction model that can gauge the satisfaction level of workers from the construction industry. This model can be extended and upgraded for other industries with some modifications to the features and the weights of the key features for job satisfaction.

## 3. Materials and Methods

In the first phase, a detailed literature review was conducted for this research. The problem was supported by additional pieces of evidence and a research gap was illustrated for this research. The research aim of this paper was aligned with the problem statement.

In the second phase, the previous decision support models were reviewed in various disciplines. The models' merits and demerits were explored and it was found that no specific model exists that integrates different key areas of job satisfaction for the construction industry. Thus, this paper focuses on a specific model for the construction industry that improves the job satisfaction of the workers. A total of 30 key factors of job satisfaction were identified through the detailed literature review and a pilot study was conducted to assess the suitability of the factors aligned with the construction industry in general. The pilot study involved senior professionals working in the construction industry. A total of 12 experts, each having more than 20 years of professional experience on mega construction industry projects, both locally and globally, were selected from the industry. After this pilot study, a total of 22 key factors were considered for the final set of questionnaires for this study.

In the third phase, a final questionnaire was devised and distributed within the construction industry to obtain the final features for the job satisfaction model. The decision support model was also statistically validated and validated by experts.

*Data Collection and Analysis*

The data of this study were gathered via a questionnaire, which was sent to construction industry experts via hard copy, email, and Google Forms. A total of 250 questionnaires were sent to experts and 220 were received successfully for this research. The questionnaire achieved a response rate of 88%, which is quite satisfactory. The data were downloaded from Google Forms and transferred to an Excel sheet with the hard copy data. The data were compiled and .csv files were generated to assess the data using SPSS 26.0 and via a descriptive assessment. The final model was generated and validated using Cronbach's alpha and R-square.

## 4. Results

Smart PLS was used to design the job satisfaction model for this study. The raw data were modeled in Smart PLS 26.0 and the construct model was created for the model of job satisfaction of construction workers, as shown in Figure 8.

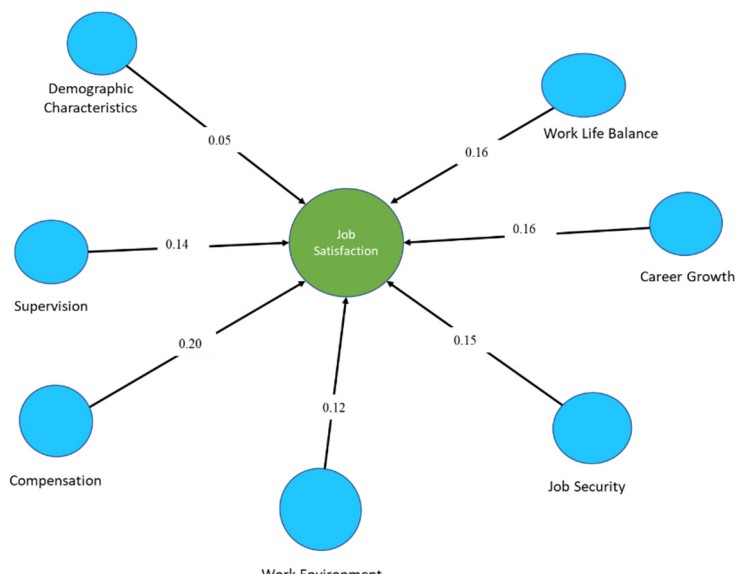

**Figure 8.** Job satisfaction model for construction workers.

Analysis showed that worker compensation has a higher weight compared to other key features, and includes the overall compensation package offered to workers by the employer. Salary, insurance, transport, residential allowance, leave, family support such as children's school fees, and medical insurance are commonly covered in the compensation package. The workers of the construction industry give this a higher priority and workers are happy at job places that offer higher compensation packages. Career growth and work–life balance have the second highest weights in the model. The workers in the construction industry feel satisfied if they feel that the company/employer offers sufficient career opportunities to grow their careers. This was observed as being one of the key features of the model because the workers want to see their chances to expand and grow their possible chances of promotion. This may include in-house promotions in the company hierarchy and the sense of satisfaction to get the next job in the case of any issue raised with an existing employer. A good and healthy work–life balance was also found to be an important feature of the job satisfaction model. Companies have different policies and statements of purpose for their employees. Furthermore, the construction industry is somewhat fragile as no specific standards are followed by all companies, except the countries' regulations, which are mostly broad and general. Fixed work timings, flexible work timings, attendance recording systems, the office governing culture, task management, deadline management, supervision procedure, family time, and reward systems are included within a healthy work–life balance.

Job security follows work–life balance in the model and is followed by supervision of the employees on the project. Construction workers feel greater job insecurity because of various reasons. Project-based tasks, different and remote work sites, tight task schedules, resource problems, and other issues can be the possible causes of insecurities. Supervision and its mechanism also play essential roles in the job satisfaction culture for the workers of the construction industry because workers feel that the culture is due to the hypocrisy of the supervising staff. A strong system should be created to resist this problem because it can potentially impair workers' productivity and satisfaction [27].

*Validation of the Model*

The developed model was run to assess the item reliability and convergent validity. For evaluation of item reliability, 0.5 was taken as the cut-off value while convergent validity parameters, i.e., CR, alpha, and AVE, were assessed using the cut-off values of 0.7 and 0.5, respectively. Variables were discarded based on the criterion that their elimination would increase the composite reliability [28]. It is suggested that the indicators with a loading <0.5 should be deleted iteratively unless CR and AVE values reach an acceptable level [29]. The results of the assessment are presented in Table 1.

**Table 1.** Convergent validity of the model.

| Construct | Cronbach's Alpha | CR | AVE |
|---|---|---|---|
| Demographic Characteristics | 0.780 | 0.849 | 0.530 |
| Supervision | 0.809 | 0.887 | 0.724 |
| Compensation | 0.875 | 0.854 | 0.597 |
| Work Environment | 0.732 | 0.880 | 0.786 |
| Work–Life Balance | 0.813 | 0.884 | 0.517 |
| Career Growth | 0.792 | 0.810 | 0.577 |
| Job Security | 0.788 | 0.779 | 0.670 |

After assessing the individual reliability and convergent validity of the measurement model, the discriminant validity of the construct was evaluated for assessing the extent to which a given construct was different from other constructs. For adequate discriminant validity, the diagonal elements need to be greater than the off-diagonal elements in the corresponding rows and columns, which is confirmed in the Fornell–Larcker criterion results shown in Table 2

**Table 2.** Fornell–Larcker criterion results of latent variable correlations.

| Fornell–Larker Criterion | DC | SP | COM | WE | WLB | CG | JS |
|---|---|---|---|---|---|---|---|
| Demographic Characteristics (DC) | **0.728** | | | | | | |
| Supervision (SP) | 0.626 | **0.809** | | | | | |
| Compensation (COM) | 0.502 | 0.526 | **0.875** | | | | |
| Work Environment (WE) | 0.450 | 0.330 | 0.344 | **0.732** | | | |
| Work–Life Balance (WLB) | 0.600 | 0.733 | 0.561 | 0.238 | **0.813** | | |
| Career Growth (CG) | 0.529 | 0.688 | 0.733 | 0.497 | 0.740 | **0.792** | |
| Job Security (JS) | 0.688 | 0.626 | 0.529 | 0.733 | 0.344 | 0.450 | **0.788** |

The criteria for evaluating the structural model include squared multiple correlations ($R^2$) and the path co-efficient ($\beta$) of each path. The results of the $R^2$ of the final model are shown in Table 3.

**Table 3.** $R^2$ score of the final model.

| Matrix | R-Square | R-Square Adjusted |
|---|---|---|
| Main | 0.802 | 0.774 |

It was found that the final model has a score of 0.802, which is within the acceptable limits of a normal model. The F-square value is the change in R-square when an exogenous variable is removed from the model. F-square measures the effect size (>= 0.02 is small; >= 0.15 is medium; >= 0.35 is large), as per [30].

In the last phase, the overall model feature reliability was also assessed using Cronbach's alpha. The overall model fit depends on the model features and the overall Cronbach's alpha fit of the model, as shown in Table 4.

**Table 4.** Cronbach's alpha of the final model.

| Model | Cronbach Score | Status |
|---|---|---|
| | 0.842 | Very Good Fit |

## 5. Conclusions

Job satisfaction, as an academic concept, has aroused wide attention from the fields of management and social psychology, and in practical operations, in recent years. Various models exist for different industries, all of which have a different level of suitability and applicability. There is no specific model that exists to gauge the satisfaction level of construction workers.

A job satisfaction model was designed using Smart PLS and SPSS in this study to assess the satisfaction level of the workers in the construction industry. It is concluded that the compensation package offered to workers in the construction industry is one of the key elements aligned with satisfaction level. The compensation package includes the salary, medical, paid leave, and family benefits provided to workers. Work–life balance is the second most important factor in the model. This covers office hours, task flexibility, timing strictness, and other sub-factors. Career growth is the third most important factor in the job satisfaction model for construction workers and includes the training and career enhancement chances of the workers and workers' expectations of growth and compensation enrichment. Job security is the fourth key factor in the model, followed by job supervision, work environment, and demographic characteristics.

Every industry has different working and career dynamics, and the construction industry is one of the most challenging industries because the job nature is not similar and the demography is also diverse. Thus, it is very important to maintain the satisfaction level of the workers in this important industry. This model can assist the stakeholders of the construction industry to analyze and assess the satisfaction level of their workers and maintain a higher satisfaction culture in their organization to improve employee retention rates and productivity.

## 6. Limitations of the Model

This model was designed for lower workers and mid-career professionals in the construction industry. The model is limited to only the main indicators (constructs) presented in this paper.

**Author Contributions:** Conceptualization, methodology, software, formal analysis, writing—original draft preparation Q.H.K.; validation, investigation, S.H.K.; writing—review, supervision N.Y.Z.; project administration and review B.S. All authors have read and agreed to the published version of the manuscript.

**Funding:** Swidweb Solutions, UTHM's spin off company, Malaysia.

**Institutional Review Board Statement:** Not Applicable.

**Informed Consent Statement:** Not Applicable.

**Data Availability Statement:** The data can be requested from the corresponding author.

**Acknowledgments:** The authors are thankful to Universiti Tun Hussain Malaysia for providing a platform to conduct this study and the authors are also grateful to Prince Sultan University Riyadh for providing scholarly support and paying the article processing charges for this paper.

**Conflicts of Interest:** The authors declare no conflict of interest.

## Nomenclature

| | |
|---|---|
| PLS | Partial least square |
| UN-SDG | United Nation Sustainability Development Goals |
| EI | Emotional intelligence |
| R&D | Research and development |
| HR | Human resource |
| SEM | Structural equation modeling |
| ISSP | International Social Survey Program |
| TIMSS | Trends in International Mathematics and Science Survey |
| HLM | Hierarchical linear modeling |
| NGO | Non-governmental organization |
| POS | Perceived Organizational Support |
| SME | Small–medium enterprises |
| CMB | Common method bias |
| CSV | Comma separated values |
| SPSS | Statistical Package for Social Sciences |
| CR | Consistency ratio |
| AVE | Average variance |

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
