# Peer review of "An Integrated Model to Improve Job Satisfaction: A Case for a Sustainable Construction Industry"

_sustainability, doi:10.3390/su15108357_

Round 1

Reviewer 1 Report

Thank you for the opportunity to review and comment on your uniquely positioned paper which evaluates different job satisfaction models to proposes an integrated job satisfaction model for construction industry workers in developing countries based on data collected from construction industry experts using a questionnaire survey with approximately 290 experts. From an academic perspective, the paper is unique in its contribution since it provides recent research insight not currently presented within academia. Equally important is the value this research provides to c-suite industry and HR executives which in turn will support strategies to support job satisfaction and motivation of construction workers. In terms of opportunities for improvement, I would recommend as author(s) you revise the current citation format from numbers to authors names and year; and ensure the reference list is structured in alphabetical order. In addition, you should note how the experts were selected and any criteria, and if experts received an incentive to participate in the research. If as author(s) you were to incorporate this data, then I believe the article would present a more comprehensive and current perspective enabling you to contribute to a refinement of the research while gaining an audience for your research.  

Author Response

Dear Sir

First of all, thank you very much for your valuable suggestions to improve the paper. We did out best to address each of your valued suggestion at our best. Please find the separate file attached with our response. 

thanks 

Reviewer 2 Report

Dear authors,  

Your paper received my best attention. Unfortunately, mainly because of conceptual and theoretical problems, I could not grasp its contribution for sustainability field. 

As a fellow researcher I know those are not easy words to read, but I encourage you to read and consider my detailed comments and use what you find suitable.

Best regards.

Comments to the Authors:

1-Being a researcher on this topic, I completely disagree with the authors when they say that “There is no model in place to measure the job satisfaction of construction industry workers”. I recognize, however, how difficult is to deal with the immense literature and different approaches to this topic. In this case, I would suggest that the authors simply choose one of the approaches and disclose their reasons for choosing that and no another branch of the literature, also focusing on sustainability as expected outcome.

2-As a matter of fact, If I may suggest something to the authors, it would be to present a clearer picture of the theoretical field from which they draw their ideas and then point what are exactly the problems that could be solved with their research (and even if those are localized managerial problems, which is something a reader might infer from reading certain statements in the paper, there must be a broader, general, interest for scholars and practitioners). As it is, the paper seems an exercise in quantitative techniques, but with no theoretical meaning.

3-As mentioned before, I do not think that the paper uses the literature properly. The main problem is that the literature review does not lead to a clear theoretical problem (and the case for the empirical research is not adequately developed).

Author Response

Dear Sir

First of all, thank you very much for your valuable suggestions to improve the paper. We did out best to address each of your valued suggestions at our best. Please find the separate file attached with our response. 

thanks 

Reviewer 3 Report

The topic of the article is very relevant and raises an extremely important problem of the value of a person as a unique creation in the modern world. It is no coincidence that the present time is called the “epoch of talentism”, since it has become obvious that the personal value of employees, their talents, aspirations, desires, abilities are paramount and human capital is the main one among other factors of production and should rightfully occupy a central place in creating and building an organization and shaping its strategies.

The unreferenced statement in the introduction needs to be corrected: " There has been a growing volume of research that supports the idea that high levels of Emotional Intelligence (EI) are a strong indicator of professional success in recent years ." However, there are no references to these works.

I propose to unite sections 3 and 4 under the general title Materials and methods. Also the sentence from section 5 “Smart PLS has been used to design the job satisfaction model for this study.” must be moved to this merged section. Sections 5 and 6 should also be merged under the heading Results.

All factors must be described: “A total number of 30 key factors of job satisfaction were identified” (Line 465) and also "a total number of 22 key factor" (Line 471).

Also, adding the survey questions (Line 473) to the appendix of the article would add value to it.

Minor remarks:

1. In the annotation it is necessary to decipher UN-SDG

2. Line 169, 172, 176, 180, 185, 362, 375, 413: Please replace "we" with "they"

3. Line 413: Please replace "our" with "their"

4. Line 278: Please instead of "[1] addresses" you need ""In the [1] authors address"

5. Line 418: Please replace "[25] conducted" with ""In the [1] authors conducted"

In general, despite the listed shortcomings, the article has undeniable strengths that allow to recommend it for publication in the Sustainability journal. First of all, it should be noted clearly built logic of the narrative. The authors adhere to the classical structure of IMRAD articles. The introduction substantiates the relevance of the study and clearly describes the scientific gap that exists in this area. An exhaustive review of the literature is given. The chronology of the development of ideas and models of the relationship between job satisfaction and organizational commitment is presented. The authors conducted a good analysis of the works, highlighting the limitations of each of them. Then they carried out their own research, covering a sufficient number of surveyed specialists in the construction industry.

I recommend that the authors, in order to give greater scientific value to the article, add a Discussion section and describe in it what are the differences between the obtained model of job satisfaction in the construction industry and other areas. In this way, you will mark the place of your research in the system of others.

Good luck with your further research!

Author Response

Dear Sir

First of all, thank you very much for your valuable suggestions to improve the paper. We did our best to address each of your valued suggestions at our best. Please find the separate file attached with our response. 

thanks 
